# *Galdieria sulphuraria* ACUF427 Freeze-Dried Biomass as Novel Biosorbent for Rare Earth Elements

**DOI:** 10.3390/microorganisms10112138

**Published:** 2022-10-28

**Authors:** Maria Palmieri, Manuela Iovinella, Seth J. Davis, Maria Rosa di Cicco, Carmine Lubritto, Marco Race, Stefania Papa, Massimiliano Fabbricino, Claudia Ciniglia

**Affiliations:** 1Department of Environmental, Biological and Pharmaceutical Sciences and Technologies, University of Caserta “L. Vanvitelli”, Via Vivaldi 43, 81100 Caserta, Italy; 2Department of Biology, University of York, Wentworth Way, York YO10 5DD, UK; 3State Key Laboratory of Crop Stress Biology, School of Life Sciences, Henan University, Kaifeng 475004, China; 4Department of Civil and Mechanical Engineering, University of Cassino and Southern Lazio, Via di Biasio 43, 03043 Cassino, Italy; 5Department of Civil, Architectural and Environmental Engineering, University of Naples Federico II, Via Claudio 21, 80125 Naples, Italy

**Keywords:** *Galdieria sulphuraria*, biosorption, rare earth

## Abstract

Rare earth elements (REEs) are essential components of modern technologies and are often challenging to acquire from natural resources. The demand for REEs is so high that there is a clear need to develop efficient and environmentally-friendly recycling methods. In the present study, freeze-dried cells of the extremophile *Galdieria sulphuraria* were employed to recover yttrium, cerium, europium, and terbium from quaternary-metal aqueous solutions. The biosorption capacity of *G. sulphuraria* freeze-dried algal biomass was tested at different pHs, contact times, and biosorbent dosages. All rare earths were biosorbed in a more efficient way by the lowest dose of biosorbent, at pH 4.5, within 30 min; the highest removal rate of cerium was recorded at acidic pH (2.5) and after a longer contact time, i.e., 360 min. This study confirms the potential of freeze-dried cells of *G. sulphuraria* as innovative ecological biosorbents in technological applications for sustainable recycling of metals from e-waste and wastewater.

## 1. Introduction

Biosorption represents an environmentally friendly, more effective, and more economical alternative to conventional metal recovery methods, i.e., pyrometallurgy and hydrometallurgy, as it uses biodegradable biosorbents that can be easily produced in high quantities [1,2]. Biosorption has been applied both for the removal of toxic metals (arsenic, chromate, cadmium, or uranium) [1] from polluted waters, and for the recovery of valuable metals such as gold, platinum, palladium, or others from solutions [2,3]. 

The main challenges of innovative technologies are material stability, selectivity, efficacy, and environmental and economic sustainability. Several studies have been carried out to overcome these challenges and most of them have focused on the use of bacterial cells, fungi, yeasts [1], algal biomass [4,5,6,7,8] or biocomponents such as crab shells [9], chicken feathers [10], etc., as biosorbents that can be easily produced or that are waste materials (e.g., in the case of crab shells) [11]. Biopolymers, on the other hand, such as cellulose, chitin, or chitosan materials are chemically resistant. However, these materials do not possess any selectivity, and bind a wide range of different elements, thus missing the selectivity factor in the metal recovery process, and limiting their potential. Biosorbents should meet specific criteria, such as: (1) high biosorption capacity; (2) suitable kinetics; (3) good size, shape, and physical properties; (4) cheap, fast, and high-performing separation of biosorbent from solutions; (5) strong mechanical strength; (6) being regeneratable and reusable [12,13,14,15,16].

*G. sulphuraria* is a polyextremophilic red alga thriving in highly acidic and thermal environments, such as volcanic environments, naturally rich in heavy, rare, and precious metals and capable of enacting unique metal tolerance mechanisms [17]. The interest in using *G. sulphuraria* living cells for bioremediation purposes [18,19] and in the bio-uptake (bioaccumulation) of metals grown very rapidly, thanks to the promising results and the increasingly comprehensive knowledge of their genomes [20,21]. Bioaccumulation is an active metabolic process which includes the interaction of metals with the cell surface (passive adsorption), followed by an active metabolism-dependent adsorption involving the transport of metal ions across the cell membrane into the cytoplasm, which is much slower than biosorption. A recent study on living cells of *G. sulphuraria* highlighted, particularly, that the adsorption of yttrium, cerium, europium, and terbium is pH-dependent [22].

In the present paper we tested the biomass of *G. sulphuraria* ACUF 427 as a biosorbent for recovery of rare earth elements from a quaternary metal system of Y^3+^, Eu^3+^, Ce^3+^, and Tb^3+^. To evaluate the effective biosorption efficiency of *G. sulphuraria* spent algal biomass and to compare with uptake, performances of living cells, different pHs, different contact times, and different biosorbent dosages were tested. Moreover, it is widely known that the biosorption can be improved by chemical modification of biomass, such as pretreatment, enhancement of the binding site, modification of binding sites, and polymerization [4,23,24]. In the present paper, acid pretreatment was employed to improve the negative charge of functional groups on the cell surface in order to magnify the removal rate.

## 2. Materials and Methods

### 2.1. Metals Stock Solutions

In this study, the removal capacity of Y^3+^, Eu^3+^, Ce^3+^, and Tb^3+^ from quaternary metal systems was analyzed using freeze-dried algal biomass of *G. sulphuraria*, at constant and equimolar concentration of 178 µmol/L. Y^3+^, Ce^3+^, Eu^3+^, and Tb^3+^ were acquired from Alfa Aesar (USA) in the form of chloride salt monohydrate (MetalCl_3_·H_2_O, 99.9%). Stock solutions were prepared by dissolving 2 grams of each metal salt in 1 liter of Milli-Q water, and acidified at pH 2.5 and 4.5 using sulfuric acid (98%). All the solutions were then sterilized with a 0.45 µm filter. To prevent interferences with the chemical analyses, all materials were rinsed with nitric acid and deionized water prior to use. In addition, the initial concentrations (C_i_) of the pH-adjusted REE solutions were verified by ICP-MS before experiments to be sure that there was no precipitation involved for the tested REE concentration (178 µmol/L). The pH of the metal solution was measured before and after sterilization (pHmeter Mettler-Toledo GmbH Process, Greifensee, Switzerland).

### 2.2. Microalgal Culture Preparation

*G. sulphuraria* ACUF427, deposited at the ACUF algal collection of the University of Naples “Federico II” (http://www.biologiavegetale.unina.it/acuf.html, accessed on 22 January 2022), was used for this study. *G. sulphuraria* ACUF427 was cultivated mixotrophically in 1-liter Erlenmeyer flasks to obtain enough biomass for the experiments, in Allen medium [25], at pH 2.0, enriched with saccharose 2 g/L and grown at 37 °C under continuous fluorescent illumination (50 µmol photons/m^2^ s) with 16/8 h light/dark, and refreshed weekly. Cell growth was determined by measuring optical density at 750 nm in a Secomam Prim light spectrophotometer. Given the exponential growth phase, the culture was centrifuged at 13,000 rpm for 5 min at 4 °C, the supernatant discarded, and the algal pellet was washed three times with Milli-Q water and freeze-dried (SP VirTis Benchmark).

### 2.3. Metal Uptake Experiments

In this study, the simultaneous removal capacity of Y^3+^, Eu^3+^, Ce^3+^, and Tb^3+^ from a quaternary metal system was studied using freeze-dried algal biomass of *G. sulphuraria* ACUF427, at constant and equimolar concentration of 178 µmol/L (Ce^3+^, 25 mg/L; Eu^3+^, 27 mg/L; Y^3+^, 16 mg/L, and Tb^3+^, 28 mg/L). The effect of pH was assessed and 2 pH values (2.5 and 4.5) were tested. The pH was measured only at the beginning and end of the experiment, as previous tests show that there is no significant change within 360 min of the experiment. The effect of biosorbent dosage was evaluated (2.5, 5, 10, and 20 mg/mL). The experiments were carried out for 360 min (time intervals 10, 30, 60, 180, and 360 min). Before adding the quaternary metal solution, the freeze-dried algal biomass was chemically pretreated. Chemical treatment was performed by mixing the freeze-dried biomass with 0.5 M sulfuric acid for 60 min on a stirring plate. Then, the biomass was recovered by centrifugation, washed to pH 7, dried again and used in the experiments. Positive controls (metal solutions without microalgal biomass) and negative controls (algal biomass without metals) were also considered. After exposure of the microalgal cells to the metal solutions for 10, 30, 60, 180, and 360 min, 2 mL of samples were taken at each time, centrifuged at 13,000 rpm for 5 min at 4 °C, and the supernatant and algal pellet were collected and analyzed for determination of REE concentrations by inductively coupled plasma mass spectrometry (ICP-MS, Aurora Bruker M90, Bremen, Germany). Samples were analyzed after digestion with 7 mL of HNO_3_ 65% and 1.5 mL of H_2_O_2_ 30% by microwave acid digestion (MARS 6 CEM).

### 2.4. Mathematical Analyses

The metal uptake capacity *Q* (mg/g) was calculated using the following equation, and then converted in µmol/g:(1)$$Q=\frac{V\left(Ci-Ce\right)}{M}\;$$

The REE removal efficiency (%) was estimated by using the following equation:(2)$$\%\;\mathrm{REE}\;\mathrm{removal}\;=\frac{\left(Ci-Ce\right)}{Ci}\times 100$$
where *C_i_*, *C_e_*, *V*, and *M* represent initial adsorbate concentration (mg/L), equilibrium adsorbate concentration (mg/L), solution volume (L), and dry weight of the biomass (g), respectively.

### 2.5. Statistical Analyses

The statistical significance of the reported adsorbent capacity at different operational conditions was judged using a two-way analysis of variance (ANOVA). Rstudio software (version 7.04, Boston, MA, USA) was used to conduct the statistical calculation at a 95% confidence interval (5% confidence level).

## 3. Results

### 3.1. Removal of Y^3+^, Ce^3+^, Eu^3+^, and Tb^3+^ in Quaternary Metal Systems at pH 2.5

The effects of biosorption parameters (pH, biosorbent dosage adsorption time) on the adsorption capacity of *G. sulphuraria* for yttrium, europium, cerium, and terbium are shown in Figure 1 and Figure 2 and the maximum adsorption capacity (µmol/g dry matter) data are shown in Table 1. The percentages were calculated as the ratio between the final and initial concentration of each metal (mg/L) in solution, regardless of the biomass. At pH 2.5, for all metals, Y^3+^, Eu^3+^, Ce^3+^, and Tb^3+^, the highest removal rates were obtained at the lowest biosorbent dosage (2.5 mg/mL) and after 360 min of contact (Figure 1). On a total removal of 850.23 ± 92.13 µmol/g, Tb^3+^ showed the highest removal uptake, corresponding to 305.27 ± 28.50 µmol/g (35.9%), followed by Ce^3+^, Eu^3+^, and Y^3+^, (Figure 1a, Table 1). Concentrations of all metals in solution dropped within the first 60 min after addition of dead cells. The measure had shown a partial release of metals into solution after 180 min followed by a new metal removal after 360 min (Figure 1a). While the metal uptake of Ce^3+^ and Eu^3+^ remained constant in the successive min, that of both Tb^3+^ and Y^3+^ significantly decreased (Figure 1a). By increasing the biosorbent dosage to 5 mg/mL, the total metal uptake significantly decreased, with a maximum recorded after 180 min (351.94 ± 28.63 µmol/g); the best uptake was again obtained with Tb^3+^ after 60 min (137.6 ± 7.5 µmol/g, Figure 1b, Table 1). 

In comparison, at the lowest biosorbent dosage, all metal ions were removed from the solution within 180 min although to a lesser extent. Similar results were obtained with 10 mg/mL; the highest metal concentration was recorded after 180 min with comparable amounts of metal uptake (Figure 1c). The increase of biosorbent dosage to 20 mg/mL caused strong reduction of metal uptake to half values (Figure 1d). Thus, in the quaternary system at pH 2.5 the element most adsorbed by *G. sulphuraria* ACUF427 was terbium, followed by cerium, europium, and yttrium (terbium > cerium > europium > yttrium).

### 3.2. Removal of Y^3+^, Ce^3+^, Eu^3+^, and Tb^3+^ in Quaternary Metal Systems at pH 4.5

Even at pH 4.5, for all metals the highest removal rates were obtained at the lowest dosage of biosorbent but in a shorter time, i.e., after 30 min of contact (Figure 2, Table 1). The total metal recovery reached 1440.50 ± 136.87 µmol/g at 2.5 mg/mL; by doubling biomass, the total metal uptake progressively halved. The highest amounts of Tb^3+^ removed (621.16 ± 5.21 µmol/g) were obtained after 30 min at 2.5 mg/mL (Figure 2a). 

The highest amounts of Eu^3+^ removed were 447.23 ± 38.51 µmol/g at the lowest biosorbent dosage and after 30 min (Figure 2a). The highest amounts removed of cerium were 214.03 ± 19.58 µmol/g at the biosorbent dosage 2.5 mg/mL (Figure 2a). Even at pH 4.5, the best removal rates were recorded at 2.5 mg/mL; terbium (621.16 ± 5.21 µmol/g of Tb^3+^) was the element most absorbed by the algae, followed by europium, yttrium, and cerium (terbium > europium > yttrium > cerium). 

### 3.3. The Effect of Variables on the Adsorption Capacity of G. sulphuraria for Rare Metals

The interactions among variables were evaluated using a 3D scatter plot surface (Rstudio) (Figure 3a–e). As shown in Table 2, *p* values of the factors biomass, pH, and time did not have the same statistical significance for each metal. For yttrium recovery, only biomass (biosorbent dosage) influenced its biosorption (*p* < 0.05) by *G. sulphuraria*; in contrast, the *p* values of pH, biosorbent dosage, and time were all lower than 0.05 for cerium, europium, and terbium, indicating that these variables significantly influenced the biosorption. The *p* values of interaction terms pH–biosorbent dosage and pH–time were both lower than 0.05 for europium and terbium, thus suggesting the influence of the interaction between these two terms on the adsorption capacity of the two metals. Cerium adsorption was affected by the interaction terms pH–time, with the *p* value being significantly lower than 0.05. 

The pH of the biosorption solution is a significant factor that influences the adsorption capacity of the biosorbent by influencing the charged conditions of the biosorbents and the ionization species of biosorbents. As pH increased, the adsorption capacity also increased. However, the increase of biosorbent dosage decreased the adsorbent capacity. As shown by the plots, the maximum adsorption capacity of *G. sulphuraria* in the first 30 min increased more than twice as the pH increased from 2.5 to 4.5. Furthermore, the influence of biosorbent dosage was as significant as that of pH.

## 4. Discussion

Few studies have been carried out on *G. sulphuraria* as a tool for biosorption of different types of metals in metal solutions. The biosorption of copper by *G. sulphuraria* was studied [26]; specifically, the interaction of copper and lead with *G. sulphuraria* in an aquatic medium was studied using stripping voltammetry. The biosorption of copper was detected using both viable and dead cells (2.4–2.8 µg/g), while lead was never biosorbed either by living and by dead cells, thus suggesting a selectivity feature of the algal strain. Freeze-dried biomass of *G. sulphuraria* was also used as a biosorbent in the recovery of Pt with an efficiency of more than 90% from a 2 M hydrochloric acid solution containing 10 mg/L Pt. In addition, Pt was adsorbed from a 2 M HCl solution along with 10 mg/L concentrations of seven metals, with a selectivity higher than that of ion exchange resins and activated carbon [27]. Freeze-dried cells of *G. sulphuraria* were shown to directly adsorb Au from simulated wastewater with a total acid concentration of 4 M, achieving an adsorption capacity of 35 ± 2.5 mg g^−1^ Au after 30 min exposure and a selectivity higher than that of an ion-exchange resin and comparable to that of activated carbon [28]. The use of *G. sulphuraria* for the biosorption of rare earth elements has already been investigated [10]. Nd(III), Dy(III), and Cu(II) were recovered (with more than 90% efficiency, at a concentration of 0.5 ppm) from a solution containing a mixture of different metals, i.e. Dy(III), Nd(III),Cu(II), Fe(II, III), Al(III), Co(II), Mn(II), Zn(II), Cr(III), and Ni(II), under semi-anaerobic heterotrophic conditions at a pH of 2.5. The removal efficiency remained unchanged at pHs in the range 1.5–2.5. Furthermore, at pH values in the range 1.0–1.5, lanthanoid ions were collected much more efficiently in the cell fractions than Cu(II) and thus successfully separated from Cu(II) dissolved in the aqueous acid. Microscopic observations showed that after biosorption, the metals in question were accumulated within the cell. 

The suitability of the Cyanidiophyceae (*G. maxima*, *G. sulphuraria*, *G. phlegrea,* and *Cyanidium caldarium*) for potential biotechnological use in the effective and innovative recovery of precious metals was successfully evaluated [29]. Specifically, their tolerance of Cl_4_K_2_Pd and AuCl_4_K was tested by monitoring the growth and metabolic response of the four different taxa, which were exposed to each of these metals at a concentration in the range 1–10 g/L. Growth was assessed 4 days (96 h) after a single metal exposure and the results were expressed as maximum growth rate (MGR). The results indicated that the precious metals can be tolerated by all the strains tested, although tolerance to Cl_4_K_2_Pd was higher than to AuCl_4_K when growth rates were considered. Biosorption represents an efficient and low-cost approach to recover lanthanides; various biomasses have been tested as biosorbents, with interesting abilities. Among macroalgae, seaweeds are considered powerful biosorbents, since they contain several binding sites such as carboxyl, amine, and hydroxyl groups able to link cations of metallic species present in solution; the authors of [30,31] evaluated the rapid and efficient biosorption capacity of *Sargassum* towards Eu, Gd, La, Nd, Pr, and Sm. Among microalgae, *Chlorella*, *Nannochloropsis*, *C. reinhardtii* (Chlorophyta), and *Microcystis* were also shown to be active biosorbents of lanthanides [32].

In this study, the removal of four rare metals from synthetic solution using freeze-dried biomass of *G. sulphuraria* as a biosorbent was investigated by assessing the effect of three parameters including pH (2.5 and 4.5), biosorbent dosage (2.5. 5, 10 and 20 mg/mL), and time (10, 30, 60, 180, and 360 min). For all the rare earths, the dose of adsorbent that performed best in terms of removal was the lowest one, i.e., 2.5 mg/mL; the best pH proved to be subneutral (4.5) and the contact time required to achieve maximum adsorption under this condition was 30 min for all the earths in the quaternary solution, except for cerium, for which the highest removal rate was recorded at acidic pH (2.5) and after a longer contact time, i.e., 360 min. Terbium was the element most easily adsorbed by the *G. sulphuraria* ACUF427. These findings agree with the results shown in the previous work [22], according to which the same *Galdieria* strain in the same quaternary system preferred terbium (in the same way as europium) and the highest removal rate was obtained at pH 4.5. A comparison between bioaccumulation [22] and biosorption of the same algal strain with the same metal system, carried out in the present study, showed comparable performance between metabolically active and inactive algae, respectively. These two processes differ in many significant points: in bioaccumulation, metals are partly bound onto the cell surface and partly imported into the cellular protoplast, with the requirements of additional energy; this mechanism can be inhibited by metabolic inhibitors such as temperature, pH, and lack of energy source; as a contrast during the biosorption process, metals interact only with the functional groups belonging to polysaccharides, lipids, and proteins on the cell surface; being biosorption metabolism-independent, the metal binding is very quick and can be achieved in a few minutes [33]. Moreover, the application of living algae can be more disadvantageous, as the process is longer than the biosorption, and the prolonged exposure to the metals may induce severe injuries to the cells, thus resulting in partial loss of cell-binding capacity and release of accumulated metal into solution [34]. The cellular response to metal exposure is influenced by the cell growth phase, as well. Therefore, using a living algal biomass, it would be difficult to make predictions about its ability to recover, unlike with a dead algal biomass whose capacity is not affected by this variable. Here, we reported the kinetics of biosorption of four rare metals and in all cases the metal binding reached the maximum rate after 30 min. 

Analysis of the data obtained in this study shows that an increase in the dose of biosorbent does not correspond to an increase in the removal rate of metals in solution. This is in line with the literature data showing the negative effect of high biosorbent dose on removal, probably due to agglomeration of the biosorbent itself, a phenomenon that reduces the available active sites used in metal–surface bonding for the biosorption mechanism [35,36,37]. In this study, it was shown that between the two pH values tested, 2.5 and 4.5, the optimal value for removal was the subneutral one. The results obtained agree with the literature, according to which the best performances of algal biomass of *G. sulphuraria* ACUF427 are registered at pH 4.5 [26]. 

Biomaterials such as freeze-dried cells of *G. sulphuraria* can be useful as ecological adsorbents, contributing to the sustainable recycling of rare and precious metals from electronic waste and wastewater rich in metal content, thus contributing to recovery from secondary sources and preventing the discharge of metal into the environment. The strength of using *G. sulphuraria* to recover rare earths is that REEs can be effectively biosorbed by *Galdieria* even if they are present in very low concentrations.

## Figures and Tables

**Figure 1 microorganisms-10-02138-f001:**
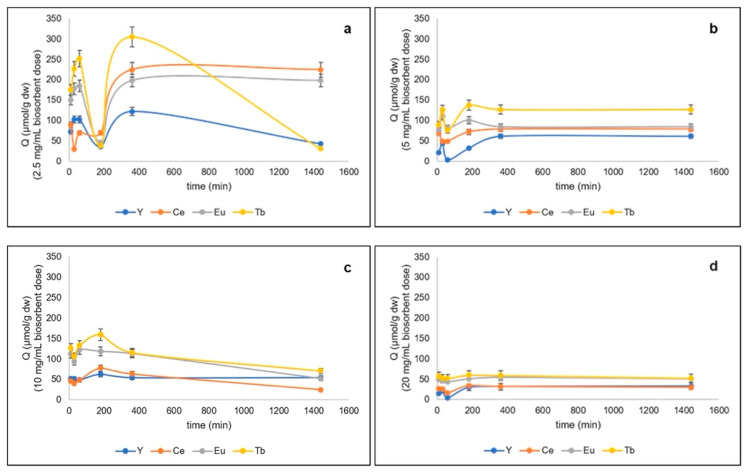
Total metals removed (µmol/g dw) from quaternary metal systems at pH 2.5 by freeze-dried algal biomass of *G. sulphuraria* ACUF427 at different doses of biosorbent: (**a**) 2.5 mg/mL, (**b**) 5 mg/ mL, (**c**) 10 mg/mL, and (**d**) 20 mg/mL after 10, 30, 60, 180, and 360 min of contact time.

**Figure 2 microorganisms-10-02138-f002:**
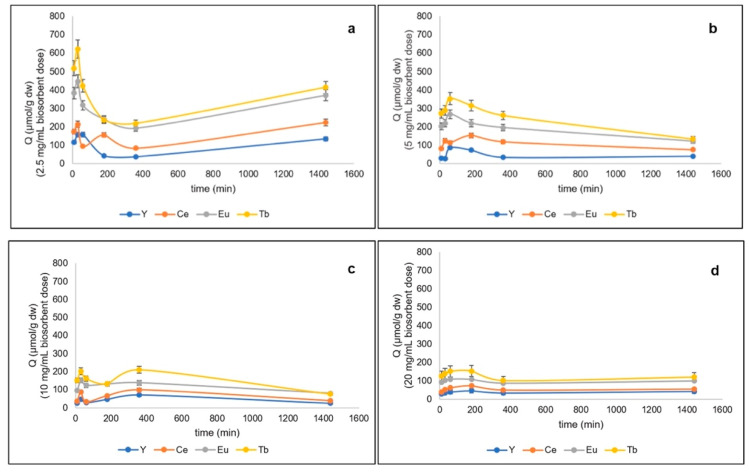
Total metals removed (µmol/g dw) from quaternary metal systems at pH 4.5 by freeze-dried algal biomass of *G. sulphuraria* ACUF427 at different doses of biosorbent: (**a**) 2.5 mg/mL, (**b**) 5 mg/ mL, (**c**) 10 mg/mL, and (**d**) 20 mg/mL after 10, 30, 60, 180, and 360 min of contact time.

**Figure 3 microorganisms-10-02138-f003:**
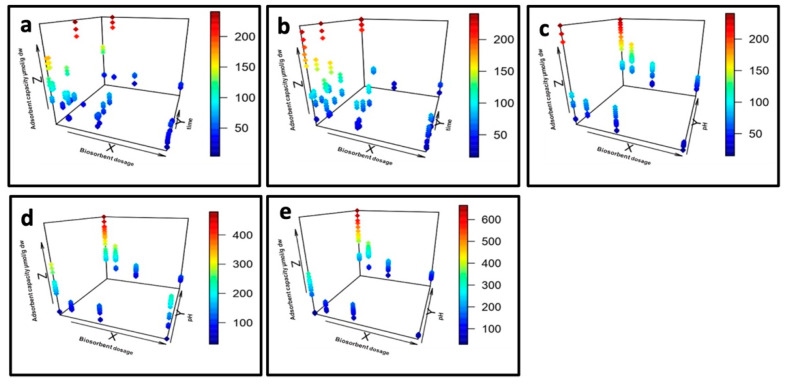
A 3D scatter plot of adsorption capacity. (**a**) Y. Effect of biosorbent dosage and time; (**b**) Ce. Effect of biosorbent dosage and time; (**c**) Ce. Effect of biosorbent dosage and pH; (**d**) Eu. Effect of biosorbent dosage and pH; (**e**) Tb. Effect of biosorbent dosage and pH.

**Table 1 microorganisms-10-02138-t001:** Maximum adsorption capacity (µmol/g dw) from quaternary metal systems at pH 2.5 and 4.5 by freeze-dried algal biomass of *G. sulphuraria* ACUF427 at biosorbent dosage 2.5 mg/mL.

	Metal	Biosorbent Dosage (mg/mL)	Time (min)	Maximum AdsorptionCapacity (µmol/g dw)
pH 2.5	Y	2.5	360	122.02 ± 13.42
	Ce	2.5	360	224.90 ± 17.60
	Eu	2.5	360	198.05 ± 15.20
	Tb	2.5	360	305.27 ± 28.50
	Total			850.24 ± 92.13
pH 4.5	Y	2.5	10	158.07 ± 22.67
	Ce	2.5	30	214.03 ± 19.58
	Eu	2.5	30	447.23 ± 38.51
	Tb	2.5	30	621.16 ± 5.21
	Total			1440.50 ± 136.87

**Table 2 microorganisms-10-02138-t002:** Analysis of variance (two way-ANOVA), regression coefficient estimates and test of significance for adsorption capacity of freeze-dried *G. sulphuraria* for yttrium, cerium, europium, and terbium. *: level of significance; * *p* < 0.05; ** *p* < 0.01; *** *p* < 0.001.

	Parameters	Df	Sum Sq	Mean Sq	*F* Value	Pr (>*F*)
Yttrium	pH	1	3635	3635	2.087	0.1509
	Biomass	1	86,458	86,458	49.637	**8.4e^−11^*****
	Time	1	5207	5207	2.989	0.0861
	pH: Biomass	1	3543	3543	2.034	0.1561
	pH: Time	1	6600	6600	3.789	0.0537
	Biomass: Time	1	201	201	0.115	0.7349
	pH: Biomass: Time	1	713	713	0.410	0.5233
	Residuals	136	236,886	1742		
Cerium	pH	1	34,038	34,038	21.004	**1.03e^−05^*****
	Biomass	1	147,468	147,468	90.999	**<2e^−16^*****
	Time	1	8846	8846	5.459	**0.02093***
	pH: Biomass	1	1146	1146	0.707	0.40187
	pH: Time	1	6931	6931	4.277	**0.04053***
	Biomass: Time	1	13,147	13,147	8.112	**0.00508****
	pH: Biomass: Time	1	5585	5585	3.447	0.06554
	Residuals	136	220,394	1621		
Europium	pH	1	125,995	125,995	28.705	**3.50e^−07^*****
	Biomass	1	185,178	185,178	42.188	**1. 44e^−09^*****
	Time	1	259	259	0.059	0.808
	pH: Biomass	1	213,672	213,672	48.680	**1.20e^−10^*****
	pH: Time	1	30,832	30,832	7.024	**0.009****
	Biomass: Time	1	2761	2761	0.629	0.429
	pH: Biomass: Time	1	40	40	0.009	0.925
	Residuals	136	596,953	4389		
Terbium	pH	1	580,111	580,111	90.781	**<2e^−16^*****
	Biomass	1	603,437	603,437	94.431	**<2e^−16^*****
	Time	1	86,475	86,475	13.532	**0.000337*****
	pH: Biomass	1	105,981	105,981	16.585	**7.86e^−05^*****
	pH: Time	1	7605	7605	1.190	0.277233
	Biomass: Time	1	22,892	22,892	3.582	0.60525
	pH: Biomass: Time	1	12	12	0.002	0.965907
	Residuals	136	869,075	6390		

## Data Availability

The data presented in this study are available on request from the corresponding author. The data are not publicly available due to privacy restrictions.

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
