# Peer review of "Galdieria sulphuraria ACUF427 Freeze-Dried Biomass as Novel Biosorbent for Rare Earth Elements"

_microorganisms, 2022, doi:10.3390/microorganisms10112138_

Round 1

Reviewer 1 Report

The use of Galdieria sulphuraria in biotechnology has been widely studied in recent years due to its resistance to extreme environmental conditions. The presented manuscript demonstrates the results of using freeze-dried algae biomass to extract rare earth metals from quaternary metal system. Undoubtedly this study confirms the potential of freeze-dried cells of G. sulphuraria as innovative ecological biosorbents for sustainable recycling of metals from e-waste and wastewater.

I have a few comments on the content of the manuscript.

The title of paper is too detailed for the title. For example, “Galdieria sulphuraria ACUF427 freeze-dried biomass as novel biosorbent for rare earth elements” would be better. The name of the algae in the title must be written in full.

Lines 88-97 A more detailed description of the medium composition with which the algae were grown and the conditions of freeze-drying is required. Was any protector used during lyophilization?

Lines 156-158 it is difficult to agree that the best recovery was for terbium (40.17%) and not for Yttrium (61.45%);

Lines 295-296 “the ideal pH proved” It is not possible to call this pH ideal since only two pH values have been studied. Perhaps pH 3.0 or 3.5 would be optimal?

Lines 326-327 these references are not listed.

The list of references is extremely sloppy.

Author Response

Dear reviewer, thank you vey much for your suggestions and comments. 

Herewith you will find my response:

-The title of paper is too detailed for the title. For example, “Galdieria sulphuraria ACUF427 freeze-dried biomass as novel biosorbent for rare earth elements” would be better. The name of the algae in the title must be written in full.

Response: we changed title: "Galdieria sulphuraria ACUF427 freeze-dried biomass as novel biosorbent for rare earth elements"

-Lines 88-97 A more detailed description of the medium composition with which the algae were grown and the conditions of freeze-drying is required. Was any protector used during lyophilization?   

Response: We did not use any protector since we were not interested to preserve viability of cells

-Lines 156-158 it is difficult to agree that the best recovery was for terbium (40.17%) and not for Yttrium (61.45%). 

Response: The percentages were calculated as the ratio between the final and initial concentration of each metal (mg/L) in solution, regardless of the biomass.

-Lines 295-296 “the ideal pH proved” It is not possible to call this pH ideal since only two pH values have been studied. Perhaps pH 3.0 or 3.5 would be optimal?

Response: We agree and we modified the text.

-Lines 326-327 these references are not listed. The list of references is extremely sloppy.

Response. All reference list has been thoroughly revised and corrected

Reviewer 2 Report

In this manuscript, the authors describe the use of G. sulphuraria ACUF427 as an absorbent for rare earth. Although the work is very interesting, below are my suggestions to make it better.

Materials and methods

-          In my opinion, it would be better if the authors always used the same unit of measurement to indicate the concentration of metals; for example, on line 76, they use µmol/L, while on line 102, they use mM; try to remain consistent throughout the text

-          Line 108-109, the authors should explain the reason for this pre-treatment and add some references

Results:

-          Remove parenthesis line 170

-          Although the results are very interesting, I find it really difficult to read the section. There are too many numbers that are also repeated in the table; I suggest rethinking this section to make it easier to read

-          It would be better to reorganize figures 3 to 7 into a single figure composed of several panels

Discussion

-          It would be interesting in this section to compare the other absorption systems available for the metals analyzed with the one presented in this study

Author Response

Dear reviewer, thank you for your suggestions and comments. We accepted all of them and improve the manuscript as follows:

-In my opinion, it would be better if the authors always used the same unit of measurement to indicate the concentration of metals; for example, on line 76, they use µmol/L, while on line 102, they use mM; try to remain consistent throughout the text.

Response: We modified throughout the text

  Line 108-109, the authors should explain the reason for this pre-treatment and add some referencesResponse: we added two references: 

Bonificio, W.D.; Clarke, D.R. Rare-Earth Separation Using Bacteria. Environ. Sci. Technol. Lett. 2016, 3, 180–184.

Mehta, S.K.; Tripathi, B.N.; Gaur, J.P. Enhanced sorption of Cu2+ and Ni2+ by acid-pretreated Chlorella vulgaris from single and binary metal solutions. J. Appl. Phycol. 2002, 14, 267–273.

  -Remove parenthesis line 170

Response: We removed the parenthesis

-          Although the results are very interesting, I find it really difficult to read the section. There are too many numbers that are also repeated in the table; I suggest rethinking this section to make it easier to read

We revised the results and rewrote in a most accessible and easier way.

-          It would be better to reorganize figures 3 to 7 into a single figure composed of several panels

Response: We reorganized figs3-7 in a single figure 

It would be interesting in this section to compare the other absorption systems available for the metals analyzed with the one presented in this study

Response: we considered more literature regarding the adsorption systems.